# The Recruitment of the Recalcitrant-Seeded *Cryptocarya alba* (Mol.) Looser, Established via Direct Seeding Is Mainly Affected by the Seed Source and Forest Cover

**DOI:** 10.3390/plants11212918

**Published:** 2022-10-29

**Authors:** Carlos R. Magni, Nicole Saavedra, Sergio E. Espinoza, Marco A. Yáñez, Iván Quiroz, Ángela Faúndez, Iván Grez, Eduardo Martinez-Herrera

**Affiliations:** 1CESAF, Facultad de Ciencias Forestales y de la Conservación de la Naturaleza, Universidad de Chile, Avenida Santa Rosa 11365, La Pintana, Santiago 8831314, Chile; 2Departamento de Ciencias Forestales, Facultad de Ciencias Agrarias y Forestales, Universidad Católica del Maule, Avenida San Miguel 3605, Talca 3466706, Chile; 3Instituto Forestal, Camino a Coronel Km 7.5, Concepción 4030000, Chile

**Keywords:** natural regeneration, recalcitrant plant species, forest restoration, provenance selection

## Abstract

Natural regeneration of recalcitrant-seeded tree species is strongly limited in Mediterranean-type climate zones due to increasing droughts imposed by climate change. Direct seeding can be a low-cost alternative to seedling establishment, but there is still limited information for some species. This study aimed to assess the effects of the seed source and forest cover on the germination and survival of the endemic *Cryptocarya alba* Mol. established through direct seeding. Three habitat types differing in forest cover were identified within the natural park Reserva Natural Altos de Cantillana, Metropolitan Region, Chile. The forest cover corresponded to open (canopy density <25%), semi-dense (canopy density around 50%), and dense forest (canopy density >75%). All forest cover had *C. alba* as one of the dominant species. At each habitat type, 38 families from four seed sources (Cuesta La Dormida (CD), Antumapu (AN), Cantillana (CA, local seed source) and Cayumanque (CY)) were directly seeded. Germination (Germin) and survival (Surv) were evaluated weekly during one growing season. There were significant differences between seed sources in Germin and Surv, with means values varying from 7.8% to 37% for Germin and 0% to 20% for Surv. The local seed source CA had the highest values in both traits. A significant variation was also observed between families within seed sources only for Germin. The dense forest cover had the highest Germin (22%) and Surv (55%) results compared to the other forest cover types, which was partially associated with differences in soil moisture, temperature, and bulk density. Due to the most frequent droughts in these Mediterranean-type climate zones, the use of local seeds on dense forest cover is recommended for the direct seeding of the species in the initial recruitment.

## 1. Introduction

There are ecological and economical concerns about the effects of more intense droughts and higher temperatures due to climate change on forests located in semi-arid Mediterranean-type climates, which encompasses a decline in forest productivity and plant species diversity and an increase in fire frequency [1]. This driest climatic scenario severely restricts the natural recruitment of recalcitrant-seeded species because they are highly sensitive to dehydration [2]. Unlike orthodox seeds, recalcitrant seeds are shed during high water content and must germinate quickly before dehydration or death [3,4]. In semi-arid Mediterranean-type climates, planting seedlings are costly and ineffective due to the low quality and availability of nursery seedlings and low success in the establishment phase [5]. In such a case, incorporating direct seeding may be a potential alternative to assist in the restoration and recovery of recalcitrant-seeded species [6,7]. High germination capacity and early survival are critical for species regeneration [8,9], but results may be highly variable because both processes are affected by the geographical origin of the genetic material [10], the microsite conditions where the mother plants grew and the restoration site [11], and even due to seed predation and herbivory [12,13]. Although our understanding of seed recalcitrance behavior has improved, especially from the point of view of species conservation [4], there is still limited knowledge about the factors influencing the germination and survival of recalcitrant-seeded species via direct seeding.

The restoration and natural recruitment of tree species are mainly limited by microsite conditions [6]. The microsite is affected by the climate and the vegetation structure, which influences the light availability, temperature, and humidity below the canopy [14,15]. In Mediterranean zones, successful restoration and higher natural recruitment are achieved under the shadows of canopies, which provide intermediate levels of light and humidity [16,17,18,19,20] and reduce weed competition. Open forests have higher soil evaporation in the upper soil layers, affecting seed germination and seedling recruitment [21,22]. Moreover, the forest cover determines the amount and type of foliage litter, which hides the seeds, decreasing their predation and slowing down their desiccation [23,24,25].

The seed source is also critical to determining the success of restoration programs and the use of local seed sources is recommended [26,27]. Local provenances may be better adapted to cope better with climatic variations in their habitat [28,29]. Nonetheless, restoration using non-local sources may be needed if local sources are not available. However, some species have limited information about the ecological limits of seed transfer. In the same way, variability within a provenance or seed source must be recognized since maternal effects influence seed germination and seedling establishment. Maternal effects are related to the seed size and seed number, dormancy, germination capacity [30], and in some cases, pathogen transmission [31]. Some forest species have shown better germination, survival, and growth rate with bigger seeds [32], whereas in other species, smaller seeds favor their dispersion [23]. Overall, seed size is related to the amount of nutrients stored [33].

*Cryptocarya alba* (Molina) Looser. (peumo) (Lauraceae family) is an endemic species from the Mediterranean-type climate zone of central Chile and is distributed from arid (31° S) to humid climates (40° S) [34]. The species is shade-tolerant and grows better in humid sites of the sclerophyllous forest type, although its development in open areas and shallow soils is mainly limited by soil moisture [35]. *C. alba* is monoecious with flowering between November and January. The fruit is a drupe that can be found in winter and spring, spread on the soil under mother plants and disappearing toward the dry summer [35]. *C. alba* seeds are recalcitrant and have exogenous dormancy due to chemical inhibitors in the pericarp and seed viability on the soil lasts five months [32]. At the nursery level, *C*. *alba* showed high phenotypic variability in the responses to water restriction [36]; however, at the landscape level, the successful establishment of the species is limited due to water scarcity [37]. In this context, there is a need for better information on how factors such as seed source and forest cover influence the recruitment of *C*. *alba* [38] under the dry conditions predicted under climate change scenarios. This information is relevant to selecting alternative restoration methods to diminish the high mortality of the establishment phase. Preliminary results suggest that sowing *C*. *alba* seeds (without pericarp) in the spring months is recommended [39], although the germination in field conditions via direct seeding is low (13% to 19%) [40]. In this study, we aimed to assess the effect of forest cover and seed source on the germination and survival of *C. alba*. The study used direct seeding to mimic the natural recruitment as an alternative for the restoration project with the species. As *C*. *alba* is a recalcitrant-seeded species, we expect a better germination capacity and seedling survival by sowing local seeds in habitats with a dense forest canopy.

## 2. Results

### 2.1. Seed Assessment in the Laboratory

Seed sources significantly differed in germination, survival, and seed mass (Table 1). The seed source CA had the highest Germin (39.4%) and Surv (14.1%), whereas seed source AN had the highest SM. We found no significant differences in H among seed sources. For all the traits, there was no observed trend associated with latitude because similar values were found for the northernmost (CD) and southernmost (CY) seed sources (Table 1). Overall, SM correlated positively with Surv (r = 0.36, *p* = 0.0318) and H (r = 0.37, *p* = 0.0604), but not with Germin (r = 0.13, *p* = 0.4455).

### 2.2. Field Experiment

There was a significant main effect of habitat type on both germination and survival (Table 2). The open and dense forest type had higher germination (around 20%) than the semi-dense type (7%). However, the results showed successful survival only in the dense forest cover (55.2%) and zero survival in the other habitat types (Table 2). Seed sources significantly differed in Germin and Surv (Table 2). Interestingly, germination means per seed source were similar to the ones obtained in the laboratory, supporting a higher germination capacity for the local seed source CA, and low germination for seed sources AN and CD (Table 1 and Table 2). Seedlings from the CA seed source were the only ones to survive after ten months. Similarly, there was a variation among mother trees within seed sources for Germin but not for Surv, with variance components and standard errors of 0.750 ± 0.251 and 0.328 ± 0.220, respectively. Regarding the maternal variation for Germin, the calculated maternal to phenotypic variance ratio (i.e., the proxy of heritability) was low 0.18.

The analysis of variance on soil moisture and temperature showed a significant effect for forest cover, date, and their interaction (*p* < 0.0001). Overall, germination peaked on October 4th when the recorded soil moisture and temperature were 0.20–0.29 m^3^ m^−3^ and 15–16 °C, respectively, and stopped on January 24th when soil moisture reached a minimum of 0.04 m^3^ m^−3^ and maximum soil temperature close to 30 °C (Figure 1). During the period, there was no clear pattern of the effect of the cover type on these soil traits, which explained the interaction between cover type and date (Figure 1). There were significant differences among forest cover on BD (*p* < 0.0010). Means and standard errors for BD for the open type were 1.01 ± 0.06 g cm^−3^, semi-dense 1.23 ± 0.07 g cm^−3^, and dense 0.74 ± 0.07 g cm^−3^.

## 3. Discussion

The lower water availability and higher temperatures due to climate change might limit the natural regeneration of some forest species of the Mediterranean-type climate zones of central Chile. This is particularly true for recalcitrant-seeded species. Our results suggest that the seed source and forest cover affect the germination of *C. alba* via direct seeding. Overall, the germination of *C. alba* was low but still higher than reported in previous studies [40,41]. Interestingly, the germination values among seed sources were similar under controlled laboratory and field conditions. This suggests that seed quality was already low among seed sources, likely influenced by the driest conditions recorded in the last years in the seed collection areas or by other environmental factors affecting mother trees during seed development [42]. In the study area, the megadrought has implied a 55 to 75% of precipitation deficit [43]. A worldwide decline in pollinators [44] has also critically impacted the Mediterranean-type ecosystem of Chile [45]. Pollen availability may affect both seed quality and germination [46]. In *C. alba*, lethal mutations in seedlings might be attributed to the low genetic variability of the species [47]. Other preliminary results in laboratory conditions with the same seed lot (not included in this study) showed that germination may be improved by removing the seeds’ pericarp of *C. alba*, reaching the best-case germination of 60%. This suggests that species have some inhibitory substances in the pericarp that constrain the germination capacity [39].

Germination peaked in October and then decreased toward mid-January. According to Figueroa [48], the germination of *C. alba* under field conditions occurs at temperatures between 10 °C and 25 °C and is inhibited over 30 °C, which agrees with our results. Similarly, germination started and peaked at soil moisture of around 0.20 m^3^ m^−3^ and decreased later toward minimum soil moisture of c.a. 0.06 m^3^ m^−3^ for these soils. Some studies showed that soil moisture is the main microsite factor affecting seed germination and establishment [49], whereas tissue desiccation is the main cause of seedling mortality when germination occurs in open areas [50]. In drier and warmer climates, as in our study, canopies may facilitate recruitment through the maintenance of higher soil moisture, lower temperatures, and herb control relative to open areas [51]. Thus, we expected a gradual increase in germination from the open to the dense forest cover, as canopies might moderate extreme environmental conditions for this shade-tolerant species, reducing seed mortality and desiccation [41]. However, the germination in our study showed no differences between the dense and open cover types, being the lowest germination in the semi-dense cover. This result might be explained by the similar soil moisture conditions during the germination period [52] or the presence of leaf litter which affects the germination of other recalcitrant-seeded species [25], but this contention needs further research.

At the end of the experiment, only the dense forest cover had some surviving seedlings, which were likely favored by the slightly higher soil moisture on this cover type during the driest months (i.e., January and February) compared with the open and semi-dense cover types. The lower light levels in this habitat type, though not measured, could have also positively influenced seedling survival. Shade-tolerant species, such as *C*. *alba*, requires shade conditions provided by dense canopy covers to regenerate successfully. Conversely, although the semi-dense forest cover had slightly higher soil moisture in spring, the germination in this cover type was probably constrained by the higher bulk density. Soil compaction has been shown to reduce seed germination, root development, and survival in forest and grass species because it reduces macroporosity, gas exchange, water flow, and root exploration [53,54]. In a laboratory study, Skinner et al. [55] found that compaction did not affect either the germination or the later mortality of *Eucalyptus albens* and *Vulpia myuros*. This result was attributed to the humid conditions of the experiment that prevented desiccation, even when radicles were not in contact with the soil surface. However, the same author mentioned that seedlings were more susceptible to surface drying in more compacted soils, which likely occurred in our study for the open and semi-dense cover types. Basset et al. [53] found that the ability of roots to penetrate the soils (e.g., root length and time to penetration) and the survival post-germination were inversely related to soil compaction, even when the range of bulk density values in their study (0.7 to 0.91 g cm^3^) were much lower than those found in the open and semi-open habitat type of our study. Overall, studies conducted in similar Mediterranean-type climate areas have reported low regeneration in woody species established in open areas [22,56]. This might have been especially critical for the recalcitrant-seeded *C. alba* because the low soil moisture condition during the growing season in the dry environment of the study site limited the early survival [57].

Variability in germination among populations was reported in other forest species and shrubs [58,59,60,61] and is attributed to genetic differences and climatic and geographic effects [62]. The Mediterranean-type climate area of central Chile has a diversity of climatic conditions due to its geography and latitudinal variation in germination capacity and early performance was observed in species of the *Nothofagus* genus [61,63,64] and sclerophyllous species including *C. alba* [37,65,66]. In our study, the success of seed germination and survival of *C. alba* was significantly determined by the seed source, where only the local seed source CA showed some survival at the end of the experiment. Seed sources are typically better adapted to their local environment; thus, restoration programs are recommended to use local seed sources [10,67]. The fact that the seed germination in laboratory and field conditions was similar for each seed source suggests a carryover effect from the mother trees. This would explain why the seed germination was largely unaltered by the environmental conditions at the study site (e.g., why the northern seed sources CD and AN did not take advantage of the milder environmental conditions at the study site). In this sense, it is common that provenances from warmer climates are adapted to longer growing found at lower altitudes but suffer from early or late frosts when moved into higher altitudes [68]. It might be possible that the opposite has occurred to seed sources CD and AN, which were moved from higher altitudes to the lower altitude of the experimental site. Additionally, our results showed a low but significant variation in seed germination among mother trees within seed sources (Appendix A), where approximately 18% of the phenotypic variance was explained by the variability of mothers within seed sources and this is the reason why we prefer using the term seed source instead of provenance. Because seeds from different mother trees were tested in a common environment, this source of variation suggests some genetic control on this trait [59]. Selection of individual genotypes within the seed source might improve germination rates of *C. alba*; however, this requires larger trials and many plots. The position of fruits within the tree crown and the age of mother trees might exert some maternal effects, influencing seed germination [69]. In recalcitrant seeds, desiccation tolerance might be an ability acquired during development before seeds are shed from the mother plants [4]. Thus, germination was also influenced by the environmental conditions experienced by mother plants in the previous generation and where the seed maturation occurs [42] which explained our results. The Mediterranean-type climate zones of central Chile have experienced a Megadrought since 2010 [43], likely affecting the maturation and quality of *C*. *alba* seeds, which need further research.

The seed mass was moderately and positively related only to survival, but not height and germination. This partially agreed with other studies on *C*. *alba* [70] and other species in Mediterranean zones [71]. Seed mass and/or size very often have significant effects on final germination percentage, germination rate, seedling survival and/or seedling growth, and even on resistance to intra- or interspecific competition; however, some studies have shown contrasting results [64,72,73]. Harper [74] suggested that the poorer performance of lighter seeds is due to their lower endosperm content. Thus, the positive correlation between seed mass and survival suggests that a higher seedling survival and growth from the larger seeds would be expected in *C*. *alba* since they contain greater amounts of nutrients than other species with tiny seeds and little reserves.

## 4. Materials and Methods

### 4.1. Seed Collection

This study utilized seeds from four seed sources of *C. alba* collected by the Centro Productor de Semillas y Árboles Forestales (CESAF), Universidad de Chile, Chile. From north to south, the seed sources corresponded to Cuesta La Dormida, Antumapu (experimental population), Cantillana, and Cayumanque, and spanned a broad range of the species distribution (Table 3, Figure 2). From May to June 2017, seeds were directly collected from 4–16 mother trees per seed source (total = 38 mothers) and then stored in sealed containers at 4 °C until experimentation. We used the International Seed Testing Association [75] standards to assess seed germination and weight in the laboratory. One hundred seeds per mother tree were weighed to calculate the individual seed mass (SM) and then used to assess germination (Germin), survival (Surv), and seedling height (H) after 45 days. H was determined with a rule. Because the species has hypogeal emergence, the germination was determined by counting the seeds with visible radicles and expressing them relative to the total seeds sowed. Survival was expressed as the surviving seedlings relative to the germinated seeds.

### 4.2. Field Experiment

A direct seeding trial was established in July 2017 in the private natural park Reserva Natural Altos de Cantillana (Cantillana seed source, Table 3). Three habitat types differing in forest cover were identified within the park and corresponded to: (1) open forest (canopy density < 25%, dominated by sclerophyllous species dominated by *Quillaja saponaria* Mol. and *C. alba*); (2) semi-dense forest (canopy density around 50%, mixed formation of *C. alba* and *Peumus boldus* Mol.); and (3) dense forest (canopy density >75%, hygrophilous forest dominated by *C. alba*, *Crinodendron patagua* Mol., and *Beilshmiedia miersii* (Gay) Kosterm). Moreover, the open forest was regenerated mainly by seeds, whereas the other cover types were by seeds and resprouting. The canopy density was determined using hemispheric photographs and the Glama software [76]. At each habitat type, a grid of 5 by 8 plots was installed, with each plot being 30 × 30 cm. Mother trees were randomly assigned to each plot and 40 seeds per mother tree were directly sowed. Prior to seeding, the litter was removed to expose the organic soil. Then, the seeds were homogeneously distributed within the plots, adding some soil and litter to avoid animal feeding [23,25]. Additionally, a 1 m height fence was built to prevent damage by herbivore mammals.

After seeding, germination was assessed weekly for the first months, whereas seedling survival was assessed during the growing season 2017–2018. Additionally, at the same dates of the germination assessment, soil moisture and temperatures were measured in the first 5 cm depth using a portable sensor Decagon GS3 (Meter Group, Inc, Pullman, WA, USA), on three measurement points per habitat type. Soil Bulk density (BD) was estimated on ten soil cores (height 5 cm, diameter 10 cm) per habitat type, which were dried after collection at 105 °C for 24 h. BD was calculated as the soil mass-to-volume ratio.

### 4.3. Statistical Analysis

We used a generalized linear model (*glm*) approach with the log-link function for binomial data (germination and survival), and traditional analysis of variance for continuous data (growth traits). Data distribution and stability of variance required for the analyses were checked graphically. The statistical model for laboratory data included only the fixed effect of seed source (levels: Cuesta La Dormida, Antumapu, Cantillana, and Cayumanque). Pearson’s coefficient of correlation was used to explore the effect of the seed weight on germination, survival, and growth. For the field experiment, the model included the fixed effects of habitat type (levels: open forest, semi-dense forest, and dense forest), seed source, and their interaction, and the random effect of the mother tree nested within the seed source. Post-hoc mean comparisons were made using Tukey’s method. Soil moisture and temperature data were modeled by a repeated measured analysis that included the fixed effects of forest cover, measurement date, and their interaction. Because these soils measurement were collected at regular intervals, the variance–covariance matrix was modeled as an autoregressive model of order 1. Differences in soil Bulk density (BD) were analyzed with habitat type as the main factor. All statistical analyses were performed using the Asreml package (VSNi, Hemel Hemepstead, UK) for R software (R Core Team, r-proyect.com, accessed on 31 July). Because of the low replication of the trials, we used a significant level of 0.1.

## 5. Conclusions

The results of this study showed that the recruitment of *C. alba* was influenced by the forest cover and seed source but not by soil conditions. Only the local seed source Cantillana showed survival post-germination, corroborating the use of local seed sources in restoration projects. Under the drier and warmer condition of Mediterranean climatic zones, the restoration via direct seeding must be mostly restricted to dense forest cover as it provides appropriate environmental conditions for seed germination and seedling establishment. However, the low rates of germination and survival of the species reflect its vulnerability to natural recruitment under the climate change scenario and support the use of alternative restoration methods such as direct seeding.

## Figures and Tables

**Figure 1 plants-11-02918-f001:**
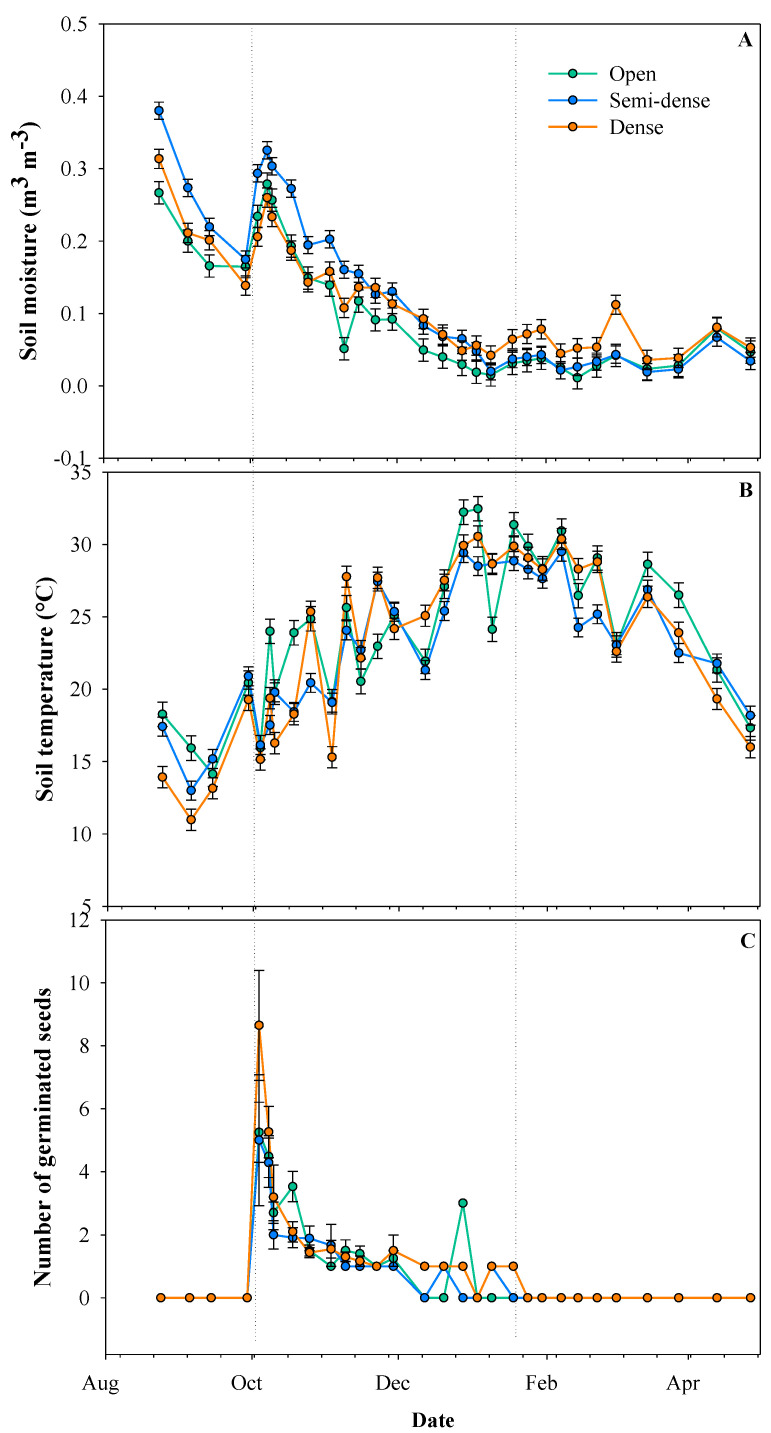
Mean soil moisture (**A**), soil temperature (**B**), and number of germinated and survived seeds (**C**) per forest cover type and date. Germination and survival were expressed based on the 40 initial seeds per mother tree. Errors bars represent the standard error.

**Figure 2 plants-11-02918-f002:**
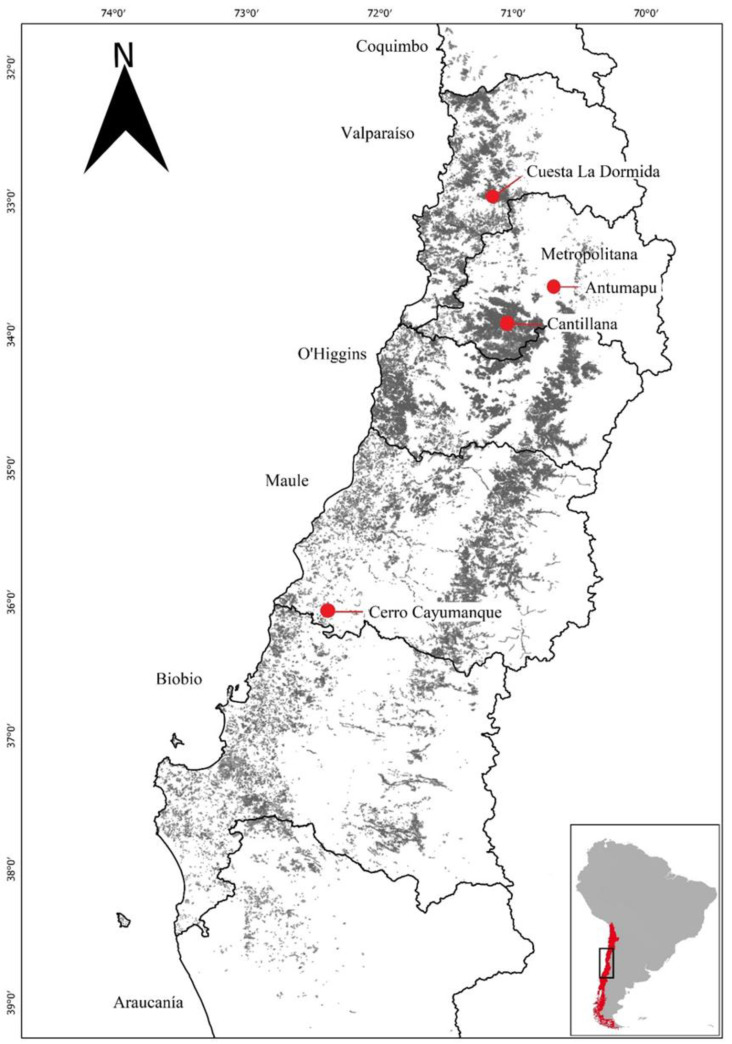
Location of the seed sources of *Cryptocarya alba* (Molina) Looser under study.

**Table 1 plants-11-02918-t001:** *p*-values from the analysis of variance and means per seed sources (plus standard errors) for germination (Germin), survival (Surv), height (H), and seed mass (SM) on *C. alba* in the laboratory experiment. Seed sources were Cuesta La Dormida (CD), Antumapu (AN), Cantillana (CA) and Cayumanque (CY). Different letters within a column indicate significant differences according to Tukey’s test.

	Germin	Surv	H	SM
***p*-values**				
Seed source	<0.0001	0.0209	0.8291	<0.0001
**Means**	(%)	(%)	(cm)	(g)
CD	10.4 ± 4.2 ^b^	3.3 ± 2.9 ^b^	5.9 ± 0.6 ^a^	0.73 ± 0.07 ^b^
AN	11.7 ± 5.2 b	10.3 ± 4.1 ^ab^	6.6 ± 0.8 ^a^	1.37 ± 0.09 ^a^
CA	39.4 ± 3.8 ^a^	14.1 ± 1.9 ^a^	5.9 ± 0.4 ^a^	1.01 ± 0.07 ^b^
CY	21.3 ± 6.9 ^ab^	4.3 ± 3.5 ^b^	6.5 ± 0.7 ^a^	0.80 ± 0.12 ^b^

**Table 2 plants-11-02918-t002:** *p*-values from the analysis of variance and means per seed source and habitat type (plus standard errors) for germination (Germin), survival (Surv) on *C. alba* in the field experiment. Seed sources were Cuesta La Dormida (CD), Antumapu (AN), Cantillana (CA) and Cayumanque (CY). Different letters within a column indicate significant differences according to Tukey’s test. Germination was determined by counting the seeds with visible radicles and expressing them relative to the total seeds sowed, whereas survival was expressed as the surviving seedlings relative to the germinated seeds.

	Germin	Surv
***p*-values**		
Forest Cover (FC)	0.0045	<0.0001
Seed source (SSo)	0.0498	0.0821
FC×Sso	0.8702	0.9734
Mother (Sso)	0.0032	0.1336
**Means per forest cover**	(%)	(%)
Open	19.0 (3.0) ^a^	0.0 (0.0)
Semi-dense	7.0 (2.1) ^b^	0.0 (0.0)
Dense	22.1 (4.2) ^a^	55.2 (5.2)
**Means per seed sources**		
CD	7.8 (2.1) ^c^	0.0 (0.0)
AN	6.5 (2.3) ^c^	0.0 (0.0)
CA	37.3 (5.4) ^a^	20.0 (3.4)
CY	19.1 (6.4) ^b^	0.0 (0.0)

**Table 3 plants-11-02918-t003:** Climatic and geographic data for seed sources Cuesta La Dormida (CD), Antumapu (AN), Cantillana (CA) and Cayumanque (CY).

Seed Sources (Number of Mother Trees)	Climatic Classification	Latitude/Longitude	Elevation (m.a.s.l.)	M.A.R. ^1^ (mm)	M.A.T. ^2^ (°C)	DM
Cuesta la dormida CD (11)	Suprathermal warm temperate, semi-arid humid regime (Csb_2_Sa)	33°04′17″/70°58′00″	743	429	14.4	17.5
Antumapu AN (7)	Temperate Mediterranean with dry hot summer (Csb)	33°34′19″/70°37′53″	629	371	14.5	15.1
Cantillana CA (16)	Suprathermal warm temperate, semi-arid humid regime (Csb_2_Sa)	33°52′07″/70°55′20″	373	543	12.0	24.7
Cayumanque CY (4)	Temperate, warm summers and cold winters (Cfb)	36°42′31″/72°31′52″	733	1292	10.0	64.6

^1^ MAR = mean annual rainfall; ^2^ MAT = mean annual temperature; DM = De Martonne aridity index, estimated as MAP/(MAT + 10).

## Data Availability

Not applicable.

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
