# Peer review of "The Recruitment of the Recalcitrant-Seeded Cryptocarya alba (Mol.) Looser, Established via Direct Seeding Is Mainly Affected by the Seed Source and Forest Cover"

_plants, 2022, doi:10.3390/plants11212918_

Round 1

Reviewer 1 Report

The paper made a field experiment to detect the effect of provenance and forest cover on seed germination and seedling survival in terms of Cryptocarya alba. The results were clear to show different results among provenances and forest cover, though the mechanisms to cause the effect were not very persuasive. As for the seedling survival, though the seedlings other than dense forest cover did not survived, and the authors suggest the dryness could be the reason for this. However, the seedlings from the provenances with drier climate also suffered high mortality, and it suggests the existence of some other factors to cause this. I think it is crucial to explain this variation, though the discussions made by the authors are not enough. Also, if the authors wanted to show the home advantage, similar experiments should be made also at other three sites.

Other questions.

L 20-21: What caused the differences in forest cover, by some forest management, or other environmental factors?

L  215-217: More detailed explanations needed: In what condition (humidity, temperature, etc.) they made in the laboratory?

L 227: If the authors wanted to verify this, the experiments should be done also in other sites where the seeds were collected.

Fig.1A: Is the differences in soil moisture enough to explain the differences in seedling survival along the forest cover? The differences seem too small.

Reviewer 2 Report

Overview

The authors describe a laboratory and a direct-seeding study in which seeds from four different provenances of C alba, a species native to Chile, were obtained and tested for germination and survival. Seeds were planted into 3 different habitat types: open forest, semi-dense forest, dense forest. They investigated germination and survival. The laboratory test revealed differences for germination and survival among provenances, with the local source (CA) having highest germination. In the field site, local seed source also had highest germination and was the only source to survive and only under dense overstory. Few studies on C alba have been published, so this is an important contribution to our understanding of the species.

Major comments

I suspect that the differences among individual (maternal) sources are amplifying provenance effects because germination rates among provenances were so variable in the laboratory. I recommend a table that shows the germination by mother tree and provenance to help us understand what is driving the observed differences. Differences among sources is commonly observed in tree improvement programs where selection is imposed to improve growth traits. Additional studies are needed to figure out the cause of poor germination in this species – potential sources include low pollination, lack of pollination vectors, and perhaps pests or pathogens. It would be interesting to study the germination rates vary over time for the same set of maternal sources to help elucidate whether low germination is truly genetic or interacts with local environmental conditions.   

Specific comments

Title. If differences in germination rates maternal sources are more pronounced than among provenances, then consider revising the title to reflect this finding. 

Introduction: please include a few sentences about the pollination complex for the species and whether the species is monoecious or dioecious. When does pollination typically occur?  Is it possible that phenology asynchrony between pollinator and flowering time may impact seed set and possibly germination rates? A recent paper did a global study on this: Rodger et al., 2021. Widespread vulnerability of flowering plant seed production to pollinator declines. Sci. Adv. 2021; 7 : eabd3524

L50-1. Wordy. Suggestion: “Although our understanding of seed recalcitrance behavior has improved…”  

L59. Delete “unlike”

L66. Remove the phrase “throughout the phenotypic plasticity.” The phrase “cope better with climatic conditions in their local habitat” basically describes the concept. 

L67-8. Delete phrase “whereas non-local provenances might confer some maladaptation to the local ones through genes flow” – gene flow is beyond the scope of this study.

L69. Instead of “seed transfer” (which is jargon) I would state “restoration using non-local sources may be needed if local sources are not available”

L77. I would also include a sentence highlighting other causes of low germination that may include (at least for other systems) a lack of pollen, lack of pollinators, asynchrony with pollinators, or biotic factors (fungi, insects). I realize there are few citations for C alba but you could refer to other systems.

L90. Replace “imposed by the climate change scenario” with “predicted under climate change scenarios”

L116. Write out “germination” and “survival” at first use. Then abbreviate other uses.

L125-126. What is more important: differences among or within provenances?  This important to explore – the variance components alone aren’t helpful without reporting total phenotypic variance which would allow you to estimate narrow-sense heritability. You could also report the CV (coefficient of variance) for provenance and source.  A table with the germination by mother tree and provenance (presumably averaged across mothers but show the number of mothers per provenance) is warranted.  

L130. Replace “were” with “where”

L138. Note that because survival was zero in the field setting for three/four provenances no p-values or Tukey’s test are needed - just report the actual finding.

L187. The relationship between germination and bulk density seems tenuous. I would think that other factors, related to the seed, would override bulk density of the soil.

L211-214. I agree that local provenances are usually better adapted to local conditions, but in your study, the differences among provenances were due to inherent differences in seed quality/germination capacity (for unknown reasons). Their failure in the field was reflective of the performance in the lab. Instead, other differences pertaining to the seed quality of non-local sources were likely at play.

L220-224. Sentences “In this sense…” and “It might be possible…” can be deleted.

L228. Replace “besides” with “within” – “Selection of individual genotypes within a provenance…”

L256. A table showing the number of families per source and the germination for each would be helpful.

Reviewer 3 Report

The manuscript presents the results of the effect of provenance on the germination and survival of the species Cryptocarya alba. The study makes a significant contribution to the recognition of the plasticity of the species in reforestation programs, which is of particular importance for C. alba and in general as a methodological approach for the conservation of recalcitrant species.

It is recognized that the manuscript has had a revision process that allows a clear and fluid reading.

In my opinion, the manuscript can be enriched by incorporating the component of the seed stands in the introduction and in the discussion, since the study determines the greater efficiency of the populations for the reforestation of this species.

Additionally, a review of the description of the data presented in Table 3 is recommended, in which it should be clarified how a sample that germinates in 22.1% survives 55.2%. In this same Table, I recommend describing how the provenance CA is the only one that shows survival with 20%, but the all provenances showed 55.2%.

Additionally, I suggest the following changes:

Page 7, Line 155, Change C alba to italics

Page 8, Lines 210 to 211, References are repeated with names and numbers. Additionally, reference 62 does not correspond to the text, which should be 60.

Page 8, Lines 257 to 261, Because low germination percentages are obtained, I suggest adding the results of viability studies by seed cutting, if they have been recorded.

Round 2

Reviewer 2 Report

Title: I think it would be more accurate to replace “provenance” with “seed source” in the title because “provenance” implies that the effects due to the location of the parents (southern vs northern sources). In your study, the “seed source,” which implies maternal effects or other factors of the seed, were stronger than provenance effects.  

L36.  I recommend revising the first sentence so it doesn’t begin with “There are concerns…”   The grammar is correct but it’s a dull opening sentence for a huge global problem. 

L37. Please define the reference to “Mediterranean” as “Mediterranean-like climate"

 L42. Delete “Usually” and revise to “Planting seedlings is often costly and ineffective…..”

L45. Delete “Securing.”  Revised sentence works without it: “High germination capacity and early survival is critical for species regeneration from seed.”

L50.  Revise to: “Although our understanding of seed recalcitrance behavior has improved…”

L54. Delete “the” before “microsite conditions”

L130. Replace “lack of survival” with “zero survival”

L134. Revise sentence starting with “unlike.” Suggestion: “Seedlings from CA provenance were the only ones to survive after xx months.”  

L135. I would remove “significant” from the sentence because you did not do a full comparison of maternal sources within provenances. 

L138. Suggested revision: “The calculated maternal to phenotypic variance ratio for germination, a proxy of heritability, was 0.18 which is a low to medium value.”  Traits such as seed size, that are under stronger genetic control, would have much higher heritability estimates than 0.18 which I consider to be on the lower end of the spectrum.  This implies that germination is influenced by maternal seed source and environment. 

L168. Revise sentence. Suggestion: “A worldwide decline of pollinators [44] has also critically impacted the Mediterranean-type ecosystem of Chile [45].”

L170. Revision suggestion: “In C. alba, lethal mutations in seedlings might be attributed to the low genetic variability of the species [47].”

L184. Replace “act facilitating” with “may facilitate”

L200. Replace “preventing” with “reducing”

L284. Replace “would have an important effect on germination” with “might improve germination rates”

L298. Awkward phrasing: “may show or not these relationships”

L302. Replace “nutritious substances” with “nutrition”

L339. Is “spinal” the correct descriptive term?  I'm not familiar with its use in this phrase. 

Fig 1. Add phrase “40 seeds per mother tree”   
